# Battery Fault: A Comprehensive Dataset and Benchmark for Battery Fault Diagnosis

**Qingdi Liu[a]**    **Yan Fu[c]**    **Lishuo Liu[b]\***    **Yanke Lin[c]\***    **Xin Jin[b]**    **Jianfeng Zhang[b]**
Chenghao Liu[c]    Lujia Pan[b]    Dongxu Guo[a]\*    Yuejiu Zheng[a]    Qiang Li[a]

[a] School of Mechanical Engineering, University of Shanghai for Science and Technology, Shanghai, 200093, China

[b] Central Research Institute, 2012 Labs, Huawei Technologies Co., Ltd., Shenzhen, 518129, China

[c] China Automotive Engineering Research Institute Co., Ltd. (CAERI), Chongqing, 401122, China

## Abstract

With the accelerated popularization of electric vehicles (EV), battery safety issues have become an important research focus. Data-driven battery fault diagnosis algorithms, built on real-world operational data, are critical methods for reducing safety risks. However, existing battery datasets have limitations such as insufficient scale, coarse-grained labels, and lack of coverage of real-world operating conditions, which seriously restrict the development of data-driven fault diagnosis algorithms. To address these issues, this paper introduces a large-scale benchmark dataset named **CH-BatteryGen**, which is, to the best of our knowledge, the first EV battery system fault diagnosis dataset based on real-world operating conditions. This dataset integrates real on-board operation data with mechanism-constrained generative modeling technology, balancing authenticity and scalability. It covers two mainstream battery chemistries, namely nickel-cobalt-manganese (NCM) lithium batteries and lithium iron phosphate (LFP) batteries, and involves charging, discharging, and operation data of 1000 electric vehicles. It provides four fault labels (normal, self-discharge, high-resistance, low-capacity) and three severity level annotations, supporting two benchmark tasks: fault classification and fault grading. Through systematic validation using traditional machine learning methods (random forest (RF), support vector machine (SVM)) and deep learning models (long short-term memory (LSTM), convolutional neural network (CNN)), the results show that the CNN model performs best in the fault classification task, achieving an F1-score of 0.9280 in the LFP discharging scenario; in the fault grading task, the F1-score reaches 0.8813. The CH-BatteryGen dataset has been open-sourced, aiming to provide a standardized evaluation platform for battery fault diagnosis algorithms, promote research development in this field, and contribute to the transformation of sustainable transportation systems.

## 1 Introduction

### 1.1 Research Background and Significance

In recent years, green transportation and low-carbon mobility have been widely recognized by governments worldwide as crucial pathways to achieving carbon peaking and carbon neutrality targets (Chen et al., 2024). Against this backdrop, electric vehicles (EVs) and energy storage systems have experienced rapid development, with their safe and reliable operation heavily dependent on lithium-ion batteries (Lai et al., 2021). However, battery faults may lead to severe consequences. For instance, internal short circuits or thermal runaway can trigger fires and even explosions, posing direct threats to human safety (Qiao et al., 2025). At the same time, battery failures can significantly

---

\*Corresponding authors. Emails: `liulishuo@huawei.com`, `linyanke@caeri.com.cn`, `guodongxu@usst.edu.cn`

impair the driving range of EVs: when the battery capacity decays below 80%, the range typically decreases by more than 20%, thereby directly affecting user experience and market acceptance. Therefore, accurate battery fault diagnosis is not only a key means of ensuring safe operation and reducing economic risks but also enables early warning to extend battery lifespan, reduce resource waste, and further promote the sustainable development of EVs and energy storage systems.

## 1.2 THIS WORK AND CONTRIBUTIONS

To address the limitations of existing battery datasets—including insufficient scale, coarse-grained labels, limited operating condition coverage, and the lack of a unified benchmarking framework—this paper proposes and constructs an AI-generated dataset for new energy vehicle power batteries, named **CH-BatteryGen**[1]. Based on this dataset, we conduct systematic studies, and the main contributions are summarized as follows:

- **Construction of a large-scale, multi-dimensional dataset:** For the first time, we integrate real on-board operational data with generative modeling methods to build a comprehensive dataset covering 1,000 EVs and two mainstream chemistries (nickel-cobalt-manganese (NCM) and lithium iron phosphate (LFP)). The dataset contains multi-dimensional time-series information such as voltage, current, temperature, and state of charge (SOC), and is annotated with four fault labels—"normal," "high internal resistance," "low capacity," and "self-discharge." This overcomes the limitations of existing public datasets that are mostly constrained to state-of-health (SOH) or binary labels.

- **Establishment of a unified benchmarking framework:** We design a multi-task benchmark system encompassing both fault classification and fault grading, systematically evaluating traditional methods such as random forest (RF) and support vector machine (SVM), as well as deep learning models such as long short-term memory (LSTM) and convolutional neural networks (CNN). Results demonstrate that while traditional methods are limited in handling complex tasks, CNN consistently achieves superior performance across different chemistries and operating conditions, exhibiting stronger robustness and generalization.

- **Revealing cross-scenario sensitivity:** Experiments show that model performance varies significantly across different chemistries (NCM and LFP) and operating modes (charging and discharging). In particular, traditional methods experience more than a 20% drop in F1-score under discharging conditions, while deep learning models exhibit smaller degradation, indicating stronger robustness to scenario changes.

In summary, the proposed CH-BatteryGen dataset and benchmark framework not only fill the gap in battery fault diagnosis data and standards but also provide a reproducible baseline for subsequent algorithm optimization and engineering applications under complex operating scenarios.

## 2 RELATED WORKS

Existing public datasets mainly focus on battery state of health (SOH) and remaining useful life (RUL), while dedicated datasets for fault diagnosis remain scarce. For example, the EVBattery dataset is built from real on-board data and contains more than 1.2 million charging segments with multi-dimensional time-series information such as voltage, current, and temperature. However, it only provides binary labels ("normal/abnormal"), which cannot capture fine-grained patterns such as lithium dendrite growth or internal short circuits (He et al., 2022). The BatteryML platform integrates 383 cycling records from seven public datasets to support model development, but its labels are limited to capacity degradation levels, without covering specific fault types (Zhang et al., 2023). The BatteryLife dataset spans multiple chemistries including zinc-ion and sodium-ion, and offers 421 charging/discharging protocols, yet annotations are restricted to a coarse-grained classification of "capacity below 80%", which is insufficient for complex diagnostic tasks (Tan et al., 2025). In addition, datasets such as the NASA battery aging archive (Saha et al., 2008) and the HNEI dataset (Devie et al., 2018) are widely used, but they suffer from large discrepancies between testing conditions and real on-board scenarios, and they lack key information such as temperature

---

[1]https://github.com/CH-BatteryGen/dataset-warehouse

distributions and cell consistency. Overall, current datasets fall short in terms of label granularity and operating condition coverage, limiting their ability to support high-precision fault diagnosis research.

At the algorithmic level, traditional approaches mainly rely on handcrafted features and machine learning models (Qiao et al., 2024b;a; Sun et al., 2022). For instance, Ren et al. (Ren et al., 2020) applied SVM to extract frequency-domain features of voltage fluctuations for high-resistance detection, achieving 92% accuracy on a laboratory dataset, but the performance dropped by more than 15% under varying temperature conditions. Xue et al. (Xue et al., 2021) proposed a random forest-based self-discharge detection method using slope variations of current–voltage curves, yet it exhibited a misclassification rate of up to 20% due to cell inconsistency. Overall, such methods strongly depend on feature engineering and lack generalization in multi-fault coupling scenarios (Huang et al., 2022). By contrast, deep learning has significantly improved diagnostic performance through automatic feature extraction. Hong et al. (Hong et al., 2019) employed LSTM to capture abnormal voltage fluctuations, enabling early warning of internal short circuits up to 50 cycles in advance with an AUROC of 89%. Ma et al. (Ma et al., 2022) combined GRU with an attention mechanism, improving the recall rate of high-resistance detection to 94%. Li et al. (Li et al., 2020) utilized a three-layer CNN to analyze surface temperature data, achieving an average of 120 seconds advance warning for thermal runaway. However, most of these studies are based on proprietary or self-collected datasets, and lack standardized experimental design, making cross-comparison among algorithms difficult. For example, Deng et al. (Deng & Hooi, 2021) achieved 71.8% AUROC with a GDN model on EVBattery, but performance dropped to 62.3% on BatteryML, highlighting the strong influence of dataset differences on algorithm evaluation.

Despite significant progress, the benchmarking system for battery fault diagnosis remains incomplete. Some studies only report overall accuracy. For example, Liu et al. (Liu et al., 2021) reported 95% accuracy in high-resistance detection, but overlooked the fact that fault samples accounted for only 5%, leading to an actual missed detection rate as high as 30%. Other studies rely excessively on idealized laboratory data. Han et al. (Han et al., 2022) validated a CNN model using noise-free voltage data, without considering sensor noise or missing data, which weakened the model's robustness in real on-board scenarios. More critically, there is currently no unified benchmarking framework for fault diagnosis. Most existing works adopt metrics inherited from SOH estimation, such as RMSE, whereas fault diagnosis requires fine-grained evaluation metrics such as confusion matrices and weighted F1-scores (Zhang et al., 2025; Li et al., 2024). In addition, the diversity of operating conditions is often neglected. For instance, Shen et al. (Shen & Kwok, 2023) reported that an LSTM model achieved an AUROC of 90% under 1C slow charging, but the performance sharply dropped to 65% under 3C fast charging, underscoring the inadequacy of condition coverage. Overall, the lack of a unified and reproducible benchmarking system has become a critical bottleneck restricting the engineering application of battery fault diagnosis (Zhou & Zhang, 2024).

## 3 DATASET DESCRIPTION

The publicly released CH-BatteryGen dataset is a generated dataset. Due to commercial confidentiality and data-security restrictions associated with real fleet data, raw fault-labeled operational telemetry cannot be directly released. Instead, large-scale real EV data are used for calibration, parameter extraction, statistical constraint modeling, and internal validation of the generation pipeline. The released dataset is generated under these physics-informed and data-constrained mechanisms, ensuring realistic operational patterns while complying with industrial data-sharing regulations. An internal realism validation protocol and distributional comparison between generated and real fleet data are provided in Appendix C.

### 3.1 DATA SOURCES

CH-BatteryGen is constructed based on large-scale real-world EV operational data, combined with AI generative models and electrochemical mechanism models to form a generation framework. The final output covers two mainstream battery chemistries, LFP and NCM. The dataset spans the full spectrum of operating states from normal to faulty, where fault modes include three representative failures: self-discharge, high internal resistance, and low capacity. Without requiring additional

preprocessing, the dataset can be directly applied to the training and validation of downstream algorithms.

Regarding the data generation methods, different strategies are adopted for LFP and NCM batteries, with detailed descriptions provided in Appendix B. A brief overview is as follows: LFP batteries are modeled using a series of multiple first-order RC equivalent circuits. The charging and discharging currents generated by Diffusion-TS (Yuan & Qiao, 2024) are used as inputs, and through the simulation of ohmic drop, polarization effects, and hysteresis characteristics, the mapping from current to voltage time-series points is realized. For NCM batteries, a discrete convolution wavelet transform (DCWT) is employed to construct the mapping model. Similarly, generated currents are used as inputs, and a three-step process—"baseline calibration, feature matrix solution, and voltage mapping"—is applied to generate voltage sequences. Leveraging the decomposition and reconstruction capabilities of DCWT, the dynamic response of voltage to current is accurately simulated (Yan et al., 2021). Compared with real testing data, the average deviation of the generated single-cell voltage is within 10 mV, with the maximum deviation not exceeding 30 mV, effectively reproducing the voltage characteristics of actual faults.

## 3.2 DATASET SCALE AND CORE FIELDS

The dataset comprises samples from 1000 EVs, including 500 vehicles equipped with NCM batteries and 500 vehicles equipped with LFP batteries. Each vehicle contains 10 charging segments and 10 discharging segments, sampled at a frequency of 10 s/point, with each segment lasting 30–60 minutes on average. The dataset covers battery packs with 28/92/96/124 cells, operating under ambient temperatures ranging from $-10°C$ to $45°C$. For each battery chemistry (NCM and LFP), the dataset contains 400 normal vehicles, 30 high-resistance fault vehicles, 30 low-capacity fault vehicles, and 40 self-discharge fault vehicles. The dataset is stored in a standardized format, with core fields accessible via mainstream analysis tools such as Python and MATLAB. The field definitions are summarized in Table 1, and representative data samples are shown in Fig. 1.

For clarity, we distinguish three levels of data granularity. A vehicle corresponds to one battery pack instance with a single assigned fault label. A segment corresponds to one continuous charging or discharging record extracted from a vehicle. Severity labels are defined only for faulty vehicles and are derived from cell-level parameter statistics within each battery pack. In segment-level experiments, individual segments are treated as samples. In vehicle-level experiments, all segments originating from the same vehicle are grouped into the same data partition to avoid potential information leakage.

Table 1: Dataset field information

| Data Field | Meaning | Precision |
|---|---|---|
| TIME | Timestamp | 1 s |
| CHARGE_STATUS | Charge status (1: charging, 3: driving/standing) | 1 |
| SUM_VOLTAGE | Total voltage [V] | 0.1 |
| SUM_CURRENT | Total current [A] | 0.1 |
| SOC | State of charge [%] | 1 |
| MAX_CELL_VOLT | Maximum cell voltage [V] | 0.001 |
| MIN_CELL_VOLT | Minimum cell voltage [V] | 0.001 |
| MAX_TEMP | Maximum temperature [°C] | 1 |
| MIN_TEMP | Minimum temperature [°C] | 1 |
| VOLT_N | Cell voltage N [V] | 0.001 |

The dataset incorporates fault samples based on well-defined electrochemical mechanisms. Different fault severity levels are evaluated by considering the 95th percentile parameters within a battery pack (e.g., the capacity/resistance of 96 cells, excluding outliers), ensuring consistency in fault definitions. The detailed classification and sample distribution are summarized in Table 2. For a comprehensive summary of dataset composition and label taxonomy, please refer to Appendix A.

Taking LFP batteries as an example, vehicle samples at the most severe fault levels under three representative fault conditions—self-discharge, high internal resistance, and low capacity—as well as the normal state were selected for comparison. The corresponding cell voltages are illustrated in Fig. 2, highlighting the distinctive features of different fault types.

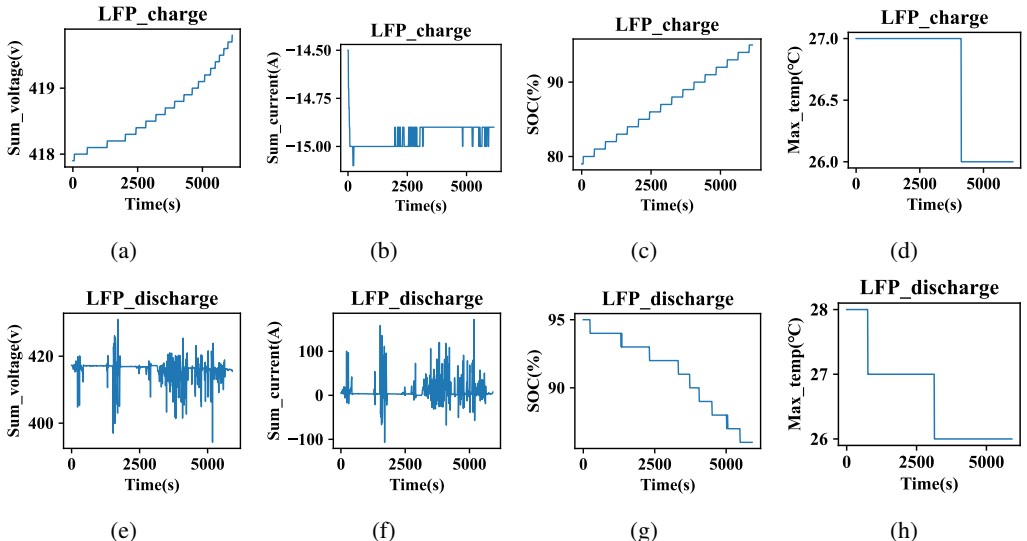

Figure 1: Display of basic battery data samples in CH-BatteryGen. Subplots (a)–(d) show the total voltage, total current, SOC, and maximum temperature during charging, while (e)–(h) illustrate the corresponding fields during discharging.

Table 2: Fault types and sample distribution.

| Fault Type | Fault Indicator | Evaluation Index | fault_index |
|---|---|---|---|
| Normal | 0 | – | – |
| Self-discharge | 1 | Mild: $0.2 \leq$ fault_index $< 1$
Moderate: $1 \leq$ fault_index $< 2$
Severe: fault_index $\geq 2$ | Leakage capacity (An/Day) |
| High resistance | 2 | Mild: $1.5 \leq$ fault_index $< 2.5$
Moderate: $2.5 \leq$ fault_index $< 3.5$
Severe: fault_index $\geq 3.5$ | $R/R_{95}$ |
| Low capacity | 3 | Mild: $0.9 \leq$ fault_index $< 0.95$
Moderate: $0.84 \leq$ fault_index $< 0.9$
Severe: fault_index $< 0.84$ | $Q/Q_{95}$ |

For normal cells (Fig. 2a, 2e), whether during charging or discharging, the voltage curves of individual cells remain highly consistent, without evident fluctuations or deviations. In contrast, cells with severe faults exhibit pronounced differences in both phases. For self-discharge faults (Fig. 2b, 2f), persistent leakage leads to a slower voltage rise during the early charging stage, accompanied by more evident curve fluctuations; during discharging, voltages drop abruptly with high-frequency oscillations. For high internal resistance faults (Fig. 2c, 2g), the increased ohmic drop produces a significantly higher charging plateau compared to normal cells under the same current; in discharging, the initial voltage drop is sharper, and the overall level stays consistently lower. For low capacity faults (Fig. 2d, 2h), the reduced effective capacity causes cells to reach the cut-off voltage earlier, resulting in shortened charging duration; during discharging, the voltage plateau is both lower and shorter in duration.

In traditional fault classification, charging data are often prioritized because the charging process is actively regulated by the battery management system (BMS), yielding more stable and controllable features. However, CH-BatteryGen also provides discharging data, which, although more complex, better reflect real-world driving scenarios. This offers crucial support for developing fault classification algorithms based on discharging conditions and fills a gap in existing battery diagnostic datasets.

## 4 BENCHMARK TASKS AND METHODS

This study designs two core benchmark tasks to address the practical requirements of battery fault diagnosis, covering the full pipeline from *fault identification* to *fault severity assessment*. These tasks aim to provide precise diagnostic insights for BMS. The first task is **fault classification**, where

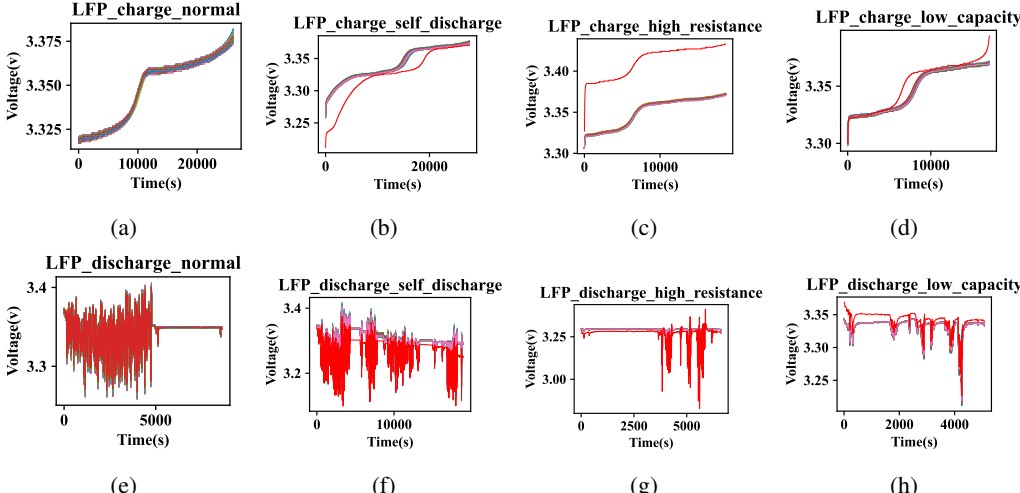

Figure 2: Comparison of charging and discharging cell voltages of LFP batteries under normal and severe fault states: (a–d) charging and (e–h) discharging processes.

the input consists of complete charging and discharging process data (including key dimensions such as cell voltage, current, and temperature), and the output comprises four classes: normal, self-discharge, high internal resistance, and low capacity. This task focuses on distinguishing between healthy and faulty batteries while further specifying the fault type, thereby supporting maintenance decision-making. The second task is **fault grading**, which targets cells already identified as faulty. Based on time-series data, the model outputs one of three severity levels—mild, moderate, or severe—facilitating repair prioritization and reducing operational risks.

To tackle the multi-scale distribution of fault features across varying pack sizes and cell counts, we propose a multi-modal benchmark model, **BatteryMultiModalCNN**. This model integrates CNN and MLP to process both image-based and numerical features. The architecture consists of four main components: image feature extraction, attention modules, numerical feature processing, and feature fusion with classification. The image feature extractor is based on a pre-trained ResNet50, where the input layer is modified to accept single-channel grayscale voltage images, while preserving residual blocks for both local and global representation learning. A CBAM attention module is embedded at higher layers to highlight channel and spatial features most relevant to faults. For numerical features, a two-layer fully connected network maps 12-dimensional statistics into a 64-dimensional representation, ensuring consistency with the image feature granularity. At the fusion stage, both feature types are concatenated and fed into a multi-layer fully connected classifier, producing either four-class or three-class outputs for classification and grading tasks, respectively.

In terms of data processing, raw time-series signals are first transformed into grayscale voltage images. To eliminate recording length differences, the time axis is normalized to the range [0,1]. Each voltage curve is then plotted into a $512 \times 512$ pixel grayscale image without axes, followed by median filtering and super-resolution enhancement to refine details. This allows fault-related voltage patterns to be more clearly visualized, as shown in Fig. 3. In parallel, statistical descriptors such as mean, standard deviation, extrema, and range are extracted from cell voltage sequences, along with consistency-based metrics. These 12-dimensional global features capture inter-cell differences and fault severity that image features alone cannot fully represent. To ensure reliable and leakage-free evaluation, we adopt two complementary data partition strategies:

(1) **Segment-level split:** individual segments are divided into training and testing sets using stratified sampling with an 8:2 ratio.

(2) **Strict vehicle-level split:** all segments originating from the same vehicle are grouped into the same partition, ensuring that no segment from a test vehicle appears in training.

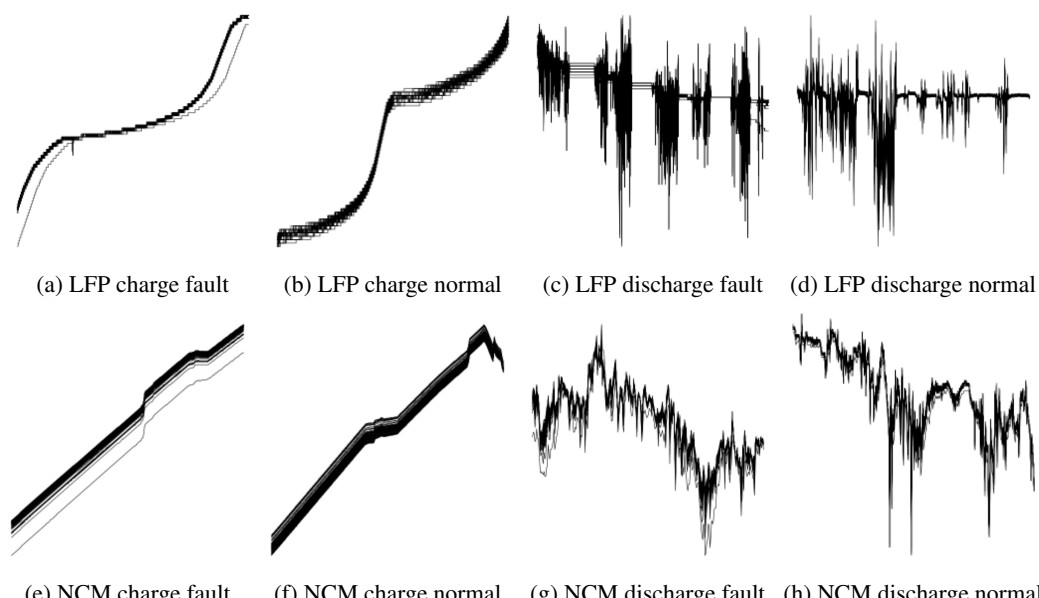

(a) LFP charge fault    (b) LFP charge normal    (c) LFP discharge fault    (d) LFP discharge normal

(e) NCM charge fault    (f) NCM charge normal    (g) NCM discharge fault    (h) NCM discharge normal

Figure 3: Examples of grayscale voltage images for fault visualization. LFP (top row) and NCM (bottom row) cells under fault and normal states during charging and discharging processes.

The vehicle-level split eliminates potential vehicle-specific pattern leakage and provides a more conservative evaluation across battery instances. All reported metrics are computed on held-out test sets under the corresponding protocol.

For evaluation, we adopt a comprehensive set of metrics, including accuracy, recall, and F1-score. Accuracy reflects prediction reliability, recall captures missed fault risks, while F1-score balances precision and recall, aligning with engineering requirements for both reliability and coverage. All metrics are computed on an independent test set to ensure objectivity and reproducibility. In experiments, we systematically compare traditional machine learning methods (e.g., RF, SVM) with deep learning models (e.g., LSTM, CNN) across image, time-series, and statistical modalities, thereby providing a thorough assessment of model adaptability and strengths across tasks.

## 5 RESULTS AND ANALYSIS

### 5.1 RESULTS OF FAULT CLASSIFICATION TASKS

At the single-file level (where a single file corresponds to one `.csv` segment storing charging/discharging data, and each generated current sequence is naturally aligned with such a segment), as shown in Table 3 and Fig. 4(a–d), traditional machine learning methods (RF, SVM) exhibit limited performance in multi-class classification, particularly under discharging conditions where all F1-scores fall below 0.71. This indicates their inability to effectively capture complex fault patterns. In contrast, deep learning methods demonstrate clear advantages: LSTM maintains relatively stable performance across different scenarios, while CNN consistently achieves the best results. Specifically, CNN reaches an F1-score of 0.9206 in the LFP discharging scenario and maintains 0.8732 in the NCM discharging scenario. The confusion matrices further reveal model-specific characteristics. In the LFP scenario, CNN achieves overall high accuracy but still shows some misclassification between "self-discharge" and "low capacity." In the NCM scenario, part of the "self-discharge" and "high resistance" samples are misclassified as normal, reflecting less distinct fault boundaries in this chemistry. Overall, CNN demonstrates superior classification accuracy and robustness across both battery chemistries and operating conditions.

At the vehicle-level, as shown in Table 3 and Fig. 4(e–h), deep learning models again outperform traditional approaches. In particular, CNN achieves F1-scores of 0.9280 and 0.8897 under LFP discharging and NCM charging scenarios, respectively, significantly surpassing other models. Most

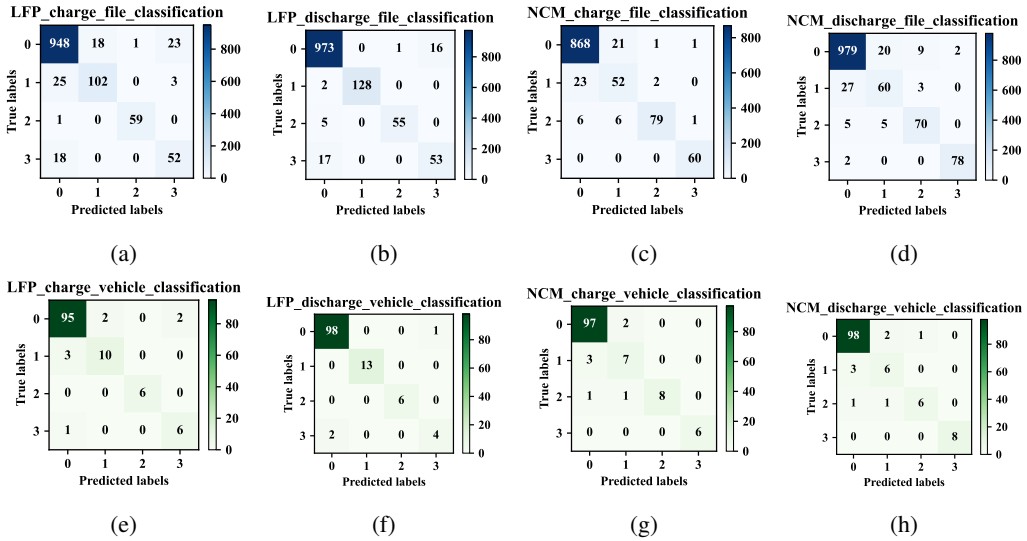

Figure 4: Confusion matrices of the CNN model for fault classification. (a–d) single-file results (e–h) single-vehicle results.

predictions are concentrated along the diagonal of the confusion matrices, though misclassification between "self-discharge" and "low capacity" still occurs under discharging and NCM charging conditions. In summary, CNN consistently delivers the highest accuracy and generalization performance in both single-file and vehicle-level fault classification tasks.

Table 3: Multi-class classification results of battery faults based on single-file and single-vehicle tasks.

| Task | Model | Dataset | Precision | Recall | F1-score |
|---|---|---|---|---|---|
| Single-file | RF | LFP_charge | 0.8600 | 0.6689 | 0.6908 |
| | | LFP_discharge | 0.7070 | 0.6677 | 0.6860 |
| | | NCM_charge | 0.7871 | 0.7454 | 0.7582 |
| | | NCM_discharge | 0.8158 | 0.6643 | 0.6808 |
| | SVM | LFP_charge | 0.6970 | 0.7199 | 0.7077 |
| | | LFP_discharge | 0.8354 | 0.5359 | 0.6222 |
| | | NCM_charge | 0.7991 | 0.8111 | 0.8033 |
| | | NCM_discharge | 0.8430 | 0.6558 | 0.7272 |
| | LSTM | LFP_charge | 0.8686 | 0.8448 | 0.8558 |
| | | LFP_discharge | 0.8724 | 0.8004 | 0.8273 |
| | | NCM_charge | 0.8721 | 0.8650 | 0.8676 |
| | | NCM_discharge | 0.8543 | 0.8260 | 0.8374 |
| | CNN | LFP_charge | 0.8639 | 0.8671 | 0.8647 |
| | | LFP_discharge | 0.9315 | 0.9103 | 0.9206 |
| | | NCM_charge | 0.8893 | 0.8771 | 0.8823 |
| | | NCM_discharge | 0.8752 | 0.8715 | 0.8732 |
| Single-vehicle | RF | LFP_charge | 0.9710 | 0.7335 | 0.7511 |
| | | LFP_discharge | 0.7290 | 0.6866 | 0.7051 |
| | | NCM_charge | 0.7326 | 0.8413 | 0.7782 |
| | | NCM_discharge | 0.7271 | 0.7500 | 0.7380 |
| | SVM | LFP_charge | 0.6036 | 0.6882 | 0.6025 |
| | | LFP_discharge | 0.7210 | 0.5897 | 0.6391 |
| | | NCM_charge | 0.8210 | 0.8651 | 0.8129 |
| | | NCM_discharge | 0.9771 | 0.7500 | 0.7935 |
| | LSTM | LFP_charge | 0.7924 | 0.7493 | 0.7685 |
| | | LFP_discharge | 0.9164 | 0.8375 | 0.8664 |
| | | NCM_charge | 0.8557 | 0.8229 | 0.8313 |
| | | NCM_discharge | 0.7121 | 0.7833 | 0.7460 |
| | CNN | LFP_charge | 0.8857 | 0.8965 | 0.8899 |
| | | LFP_discharge | 0.9450 | 0.9141 | 0.9280 |
| | | NCM_charge | 0.9151 | 0.8699 | 0.8897 |
| | | NCM_discharge | 0.8711 | 0.8467 | 0.8580 |

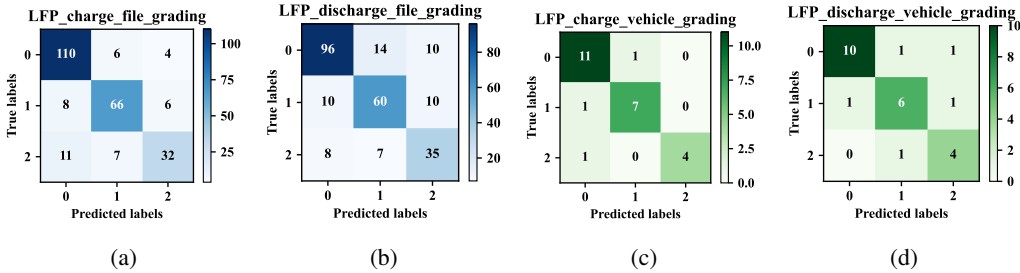

Figure 5: Confusion matrices of the CNN model for fault grading. (a–b) single-file results (c–d) single-vehicle results.

## 5.2 RESULTS OF FAULT GRADING TASKS

Compared with the classification task, fault grading is overall more challenging. In the single-file task (Table 4), traditional methods generally perform poorly, with most F1-scores below 0.65. In contrast, deep learning methods achieve significantly better results. LSTM reaches an F1-score of 0.7289 under the LFP charging condition, while CNN consistently delivers the best performance, achieving 0.8031 and 0.7442 in the LFP charging and discharging scenarios, respectively. As shown in Fig. 5(a-b), CNN can effectively distinguish different severity levels, though confusion between mild and moderate faults remains, indicating that noise under discharging conditions still poses challenges for fine-grained fault recognition.

At the vehicle-level task (Table 4 and Fig. 5(c-d)), the results are consistent with the single-file analysis: traditional methods remain insufficient, whereas deep learning methods show clear advantages. In particular, CNN achieves an F1-score as high as 0.8813 under LFP charging conditions and maintains 0.7823 under discharging conditions. The confusion matrices further illustrate that grading results are more distinct in charging scenarios, while mild and moderate faults remain more difficult to separate during discharging.

Overall, CNN demonstrates the highest accuracy and robustness in fault grading tasks. However, the difficulty of fine-grained diagnosis is notably greater than that of classification tasks. Under complex operating conditions such as discharging, mild and moderate faults are more prone to confusion due to noise, highlighting the limitations of current models in fine-grained feature extraction. Future research should focus on improving recognition capability under complex conditions.

Table 4: Multi-class grading results of battery faults based on single-file and single-vehicle tasks.

| Task | Model | Dataset | Precision | Recall | F1-score |
|---|---|---|---|---|---|
| Single-file | RF | LFP_charge | 0.6317 | 0.6400 | 0.5976 |
| | | LFP_discharge | 0.5238 | 0.5360 | 0.5288 |
| | SVM | LFP_charge | 0.6469 | 0.6440 | 0.6419 |
| | | LFP_discharge | 0.5420 | 0.5280 | 0.5323 |
| | LSTM | LFP_charge | 0.6777 | 0.7885 | 0.7289 |
| | | LFP_discharge | 0.7087 | 0.7019 | 0.7053 |
| | CNN | LFP_charge | 0.8167 | 0.7939 | 0.8031 |
| | | LFP_discharge | 0.7397 | 0.7500 | 0.7442 |
| Single-vehicle | RF | LFP_charge | 0.5465 | 0.6800 | 0.6050 |
| | | LFP_discharge | 0.6280 | 0.6400 | 0.6171 |
| | SVM | LFP_charge | 0.8056 | 0.6611 | 0.6874 |
| | | LFP_discharge | 0.6732 | 0.6000 | 0.5441 |
| | LSTM | LFP_charge | 0.8000 | 0.6667 | 0.7273 |
| | | LFP_discharge | 0.7556 | 0.7000 | 0.7205 |
| | CNN | LFP_charge | 0.9071 | 0.8639 | 0.8813 |
| | | LFP_discharge | 0.7753 | 0.7944 | 0.7823 |

## 6 CONCLUSION

This paper proposes and constructs **CH-BatteryGen**, a large-scale benchmark dataset for battery fault diagnosis. Compared with existing public datasets, CH-BatteryGen demonstrates significant advantages in scale, label diversity, and task coverage. By integrating real onboard operational data

with generative augmentation methods, the dataset achieves both authenticity and scalability. It provides multi-label annotations covering four typical fault types—normal, high resistance, low capacity, and self-discharge—and establishes a unified benchmarking framework that supports multiple tasks, including fault detection, classification, and grading.

Experimental results show that traditional machine learning methods are limited in handling complex multi-class tasks, whereas deep learning models achieve overall superior performance. In particular, CNN consistently yields the best results across different chemistries and operating conditions, maintaining high accuracy and robustness even under challenging discharging scenarios. Furthermore, the experiments reveal clear performance sensitivity to operating conditions and battery chemistries: traditional methods suffer substantial degradation under discharging scenarios, whereas deep learning models degrade less severely, demonstrating stronger robustness and adaptability.

Nevertheless, fault grading tasks remain more difficult than classification tasks, especially under discharging conditions where mild and moderate faults are easily confused, indicating room for improvement in fine-grained diagnosis. Moreover, limitations still exist in CH-BatteryGen, such as restricted chemistry coverage, limited fault label diversity, and insufficient data for extreme operating conditions, which highlight directions for future dataset expansion and optimization.

In summary, CH-BatteryGen provides the first systematic large-scale benchmark platform for intelligent battery diagnosis. It establishes a foundation for fair comparisons and reproducible research across multiple algorithms, tasks, and scenarios. We expect CH-BatteryGen to accelerate the standardization and engineering applications of battery intelligent diagnosis, while offering new opportunities for fine-grained fault recognition under complex operating conditions.

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

SUPPLEMENTARY MATERIALS

In this section, we provide supplementary materials to complement the main paper:

- **Appendix A:** Data Card for CH-BatteryGen
- **Appendix B:** Dataset Generation Methods
- **Appendix C:** Realism Validation of Generated Data

## A  DATA CARD FOR CH-BATTERYGEN

### A.1  DATASET SUMMARY

- Total vehicles: 1000
- Total segments: 19,478 segments
- Sampling interval: 10 s
- Segment duration: 30–60 minutes
- Operating temperature range: $-10°C$ to $45°C$

Table 5: Dataset summary.

| Chemistry | Vehicles | Segment Type | Total Segments | Avg. Seg/Veh |
|-----------|----------|--------------|----------------|--------------|
| **LFP** | 500 | Charge | 5,000 | 10 |
| | | Discharge | 5,000 | 10 |
| **NCM** | 500 | Charge | 4,478 | 8.956 |
| | | Discharge | 5,000 | 10 |
| **Total** | **1,000** | **–** | **19,478** | **19.478** |

### A.2  LABEL TAXONOMY

- Fault type (Table 6): Normal, Self-discharge, High resistance, Low capacity
- Severity grading (For faulty vehicles of LFP only, Table 7): Mild, Moderate, Severe

It is important to note that the multi-level severity grading task is specifically defined for the LFP chemistry subset. Given that severe battery degradation is statistically rare in actual vehicle operations, our dataset reflects this sparsity. As shown in Table 7, the sample counts decrease as severity increases: Mild faults constitute the majority, followed by Moderate, while Severe cases are the least frequent. The specific criteria for these severity thresholds are detailed in Table 2. This deliberate imbalance challenges algorithms to detect incipient, low-magnitude anomalies, a critical requirement for early-warning systems.

Table 6: Distribution of fault types.

| Category | Fault Type | LFP (Vehs) | NCM (Vehs) | Total(Vehs) |
|----------|-----------|------------|------------|-------------|
| **Normal** | Normal | 400 | 400 | 800 |
| **Faulty** | Self-discharge | 40 | 40 | 80 |
| | High Resistance | 30 | 30 | 60 |
| | Low Capacity | 30 | 30 | 60 |
| **Total** | | **500** | **500** | **1,000** |

Table 7: Distribution of Severity grading for faulty vehicles of LFP only.

| Category | Severity grading | LFP (Vehs) |
|---|---|---|
| **Self-discharge** | Mild | 32 |
| | Moderate | 4 |
| | Severe | 4 |
| **High Resistance** | Mild | 10 |
| | Moderate | 16 |
| | Severe | 4 |
| **Low Capacity** | Mild | 10 |
| | Moderate | 10 |
| | Severe | 10 |
| **Total** | | **100** |

# B DATA GENERATION METHODS FOR CH-BATTERYGEN

## B.1 DATA GENERATION PIPELINE

The CH-BatteryGen dataset is synthesized through a multi-stage pipeline that integrates deep generative modeling with physical domain knowledge. As summarized in Table 1, all signals in the released dataset are generated to preserve privacy while maintaining statistical fidelity.

The generation process initiates with a Diffusion-TS Model trained on real-world current profiles during both charging and discharging. These current profiles are then fed into chemistry-specific (LFP or NCM) models, which are detailed in B.3 and B.4. The battery model simulate the electrochemical response to produce individual cell voltages. Finally, integrated calculation modules derive pack-level attributes detailed in B.5, including Total Voltage, SOC, Maximum/Minimum Cell Voltages, Maximum/Minimum Temperatures, ensuring that all signal dimensions remain coupled through rigorous physical constraints.

## B.2 CORE INPUT: GENERATION OF CHARGING AND DISCHARGING CURRENT SEQUENCES

The current sequence serves as the key input for voltage generation, and its fidelity directly determines the reliability of subsequent data. The dataset construction adopts the Diffusion-TS framework based on probabilistic diffusion models to generate charging and discharging currents. By combining "time-domain decomposition" and "frequency-domain denoising," the framework reconstructs realistic current dynamics. In the forward process, random Gaussian noise is gradually injected into the original EV current sequence until the signal is completely degraded; in the reverse process, the Transformer network progressively restores the true current profile from the noisy signal through denoising and decoding. Meanwhile, a time-series decomposition module is introduced to separate long-term trends (e.g., monotonic decrease in charging current during the later stages) and periodic fluctuations (e.g., small-amplitude ripples), which are further refined using multi-level wavelet packet decomposition. This ensures that the frequency characteristics of the generated currents remain consistent with those of real data, thereby avoiding unphysical high-frequency distortions.

The final generated current sequence $I_{\text{gen}}$ exhibits stable temporal resolution, with a sampling frequency set to 10 Hz and a segment length ranging from 30–60 minutes. This setup realistically covers typical driving and charging scenarios, such as *constant-speed driving*, *rapid acceleration*, and *stop-and-go conditions*, as well as mainstream slow-charging and fast-charging conditions, thereby providing reliable current inputs for subsequent voltage generation of both LFP and NCM batteries.

## B.3 LFP VOLTAGE GENERATION

For LFP batteries, a series of first-order RC equivalent circuit models is employed. The generated charging and discharging current $I_{\text{gen}}$ from Diffusion-TS is used as input, and the model simulates

internal ohmic drop, polarization effects, and hysteresis characteristics to achieve precise mapping from current to voltage time-series data, as shown in Eq. 1:

$$U = OCV(SOC) + I_{\text{gen}}R_0 + I_{\text{gen}}R_1 \left(1 - e^{-\frac{t}{\tau_1}}\right) + h \tag{1}$$

Here, $U$ represents the terminal voltage; $OCV(SOC)$ is the open-circuit voltage as a function of SOC; $I_{\text{gen}}$ is the model input current; $R_0$ is the ohmic resistance; $R_1$ is the polarization resistance; $\tau_1$ is the time constant; $t$ is time; and $h$ denotes the voltage offset term. By tuning these key equivalent-circuit parameters, the model can accurately inject fault signatures such as high internal resistance, low capacity, and self-discharge, ensuring consistency with real-world LFP battery fault behaviors.

## B.4 NCM Voltage Generation

For NCM batteries, a DCWT-based mapping model is constructed. Given the generated input current $I_{\text{gen}}$, the model applies a three-step procedure—baseline calibration, feature matrix solution, and voltage reconstruction—to produce the voltage time-series data of NCM batteries. By leveraging the decomposition and reconstruction capability of DCWT, the dynamic response of voltage to current is accurately simulated.

To ensure stable voltage computation, a baseline voltage $U_{\text{ref}}$ is introduced. Using semi-annual real NCM battery test data, a reference feature matrix $G_{\text{ref}}$ (with dimension $1 \times 4$) is derived to capture the average response characteristics of unit-cell voltage to current, as expressed in Eq. 2:

$$U_{m,hy} = G_{\text{ref}} \cdot I_{m,hy} \tag{2}$$

where $U_{m,hy}$ denotes the voltage of unit $m$ at time $hy$, and $I_{m,hy}$ is the corresponding current sequence. Substituting $I_{\text{gen}}$ into Eq. 2 yields the baseline voltage $U_{\text{ref}}$, as shown in Eq. 3:

$$U_{\text{ref}} = G_{\text{ref}} \cdot I_{\text{gen}} \tag{3}$$

For each generated segment, DCWT is employed to approximate the influence of current on voltage, and a feature response matrix $F \in \mathbb{R}^{n \times 4}$ is derived, where $n$ denotes the number of battery cells. This matrix encodes the deviation of unit-cell voltages from the baseline response, as given by Eq. 4:

$$U_{\text{origin}} - U_{\text{ref}} = F \cdot I_{\text{origin}} \tag{4}$$

where $U_{\text{origin}}$ represents the measured NCM cell voltages and $I_{\text{origin}}$ is the corresponding measured current. Finally, combining $G_{\text{ref}}$ with the generated input $I_{\text{gen}}$, the reconstructed voltage sequence $U_{\text{gen}}$ is obtained, as shown in Eq. 5:

$$U_{\text{gen}} = F \cdot I_{\text{gen}} + U_{\text{ref}} \tag{5}$$

By adjusting the dimensional response of $F$ to target faulty cells, fault signatures of high resistance, low capacity, and self-discharge can be precisely injected. The final generated NCM voltages exhibit close agreement with experimental measurements, with an average deviation of less than 10 mV and a maximum deviation of 30–50 mV, thereby faithfully reproducing the voltage characteristics of real battery faults.

## B.5 Pack level Generation

Building upon the individual cell responses from the battery model, the pack-level voltage attributes are derived to simulate realistic bus-level telemetry. The Total Voltage ($U_{total}$) is calculated as the instantaneous sum of all series-connected cell voltages, reflecting the overall potential of the battery string. To represent cell-to-cell variations and potential fault signatures, we explicitly extract the Maximum and Minimum Cell Voltages ($U_{max}, U_{min}$) at each time step:

$$U_{total}(t) = \sum_{i=1}^{N} U_i(t) \tag{6}$$

$$U_{max}(t) = \max_{i \in 1,...,N} U_i(t), \quad U_{min}(t) = \min_{i \in 1,...,N} U_i(t) \tag{7}$$

To ensure the physical consistency of the generated telemetry, the Pack-level SOC is determined by the short-board effect (minimum cell principle). Specifically, we define the pack SOC as the ratio of the minimum remaining capacity among all constituent cells to the total nominal capacity of the pack:

$$SOC_{pack}(t) = \frac{\min_{i=1}^{N}(SOC_i(t) \cdot Q_i)}{Q_{total}} \tag{8}$$

This formulation ensures that when a low capacity fault is injected into a specific cell ($Q_{cell,i} \downarrow$), the resulting pack-level SOC trajectory accurately reflects the diminished energy availability, aligning the generated data with real-world BMS behavior during cycling.

To provide a multi-modal perspective, we incorporate a lumped-parameter thermal model. The temperature evolution of each cell $T_i$ is governed by the heat generation-dissipation balance:

$$mC_p \frac{dT_i(t)}{dt} = I(t)^2 R_i - h_i A(T_i(t) - T_{amb}) \tag{9}$$

where $I(t)^2 R_i$ denotes the Joule heating and $h_i$ is the convective heat dissipation coefficient. Given the current simplified parameterization of the thermal environment, we provide the Maximum and Minimum Temperatures as reference features:

$$T_{max}(t) = \max_{i \in 1,...,N} T_i(t), \quad T_{min}(t) = \min_{i \in 1,...,N} T_i(t) \tag{10}$$

It should be emphasized that while the electrical signals ($U, I, SOC$) are highly accurate for diagnostic benchmarking, the current temperature data ($T_{max}, T_{min}$) serves primarily as a qualitative reference. Due to the simplified thermal parameters, these signals reflect physical trends (e.g., heat accumulation during high-current events) rather than absolute high-fidelity thermal distributions. We aim to enhance the thermal fidelity in future iterations.

## C    REALISM VALIDATION OF GENERATED DATA

### C.1    VALIDATION PROTOCOL

To assess the distributional alignment between generated data and real operational signals, we conduct an internal embedding-based validation. Real fleet samples (not publicly released due to confidentiality constraints) and generated samples are projected into a shared feature space using identical feature representations. Two-dimensional t-SNE embeddings are then computed to examine distributional separability.

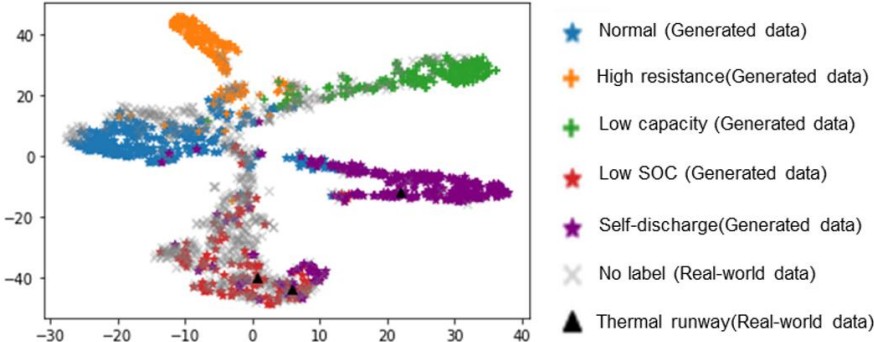

Figure 6: t-SNE visualization of real-world and generated samples. Colored points denote generated samples, including normal operation and multiple fault types (low capacity, high resistance, and self-discharge). Gray points represent real-world fleet samples used for internal validation but not publicly released. Black points correspond to thermal runaway cases observed in real-world data.

## C.2  RESULTS

As shown in Fig. 6, real and generated samples are intermingled in the embedded space without forming separable clusters, while different fault categories remain distinguishable. This indicates that the released dataset preserves the structural and fault-dependent characteristics observed in real-world data. In particular, classifiers operating in this representation space are unable to distinguish generated samples from real ones, whereas fault modes remain separable.

## C.3  INTERPRETATION AND SCOPE

We emphasize that this validation does not claim exact equivalence between the generated dataset and raw fleet telemetry. Instead, it demonstrates that the released dataset maintains internal distributional consistency and fault separability under a common representation space, while complying with data-security and confidentiality constraints.

Because the generated data preserve essential characteristics of real-world fault behaviors, models developed using CH-BatteryGen can serve as effective pretraining or benchmarking tools and can be adapted to practical diagnostic scenarios through domain-specific fine-tuning. By releasing this dataset, we aim to facilitate broader participation from the machine learning community and to promote reproducible research on battery safety monitoring.

