# OpenReview forum: "Battery Fault: A Comprehensive Dataset and Benchmark for Battery Fault Diagnosis"
_ICLR.cc/2026/Conference — ICLR 2026 Poster_

### Official Review · Reviewer_D6gL · 2025-10-30

**Soundness:** 2
**Presentation:** 1
**Contribution:** 2
**Rating:** 2
**Confidence:** 3

**Summary:**

In this paper, a new benchmark (real-calibrated, synthetically generated) for battery fault diagnosis is presented. The authors use real data for calibrating their data generation method for two different battery chemistries: lithium iron phosphate (LFP) and nickel-cobalt manganese (NCM). They use this benchmark to evaluate different baseline architectures on two tasks: fault classification and fault grading.

**Strengths:**

- Novel benchmarks that are easily accessible online are an important contribution to the academic community.
- Claimed open release in a standard format allows for comparisons in the community

**Weaknesses:**

- My main criticism is that ICLR may not be the right venue for this work. Given its specialized nature concerning batteries, I believe more specialized journals (such as those heavily cited in the references) would be a better fit.
- The data generation pipeline is quite unclear. On one hand, it states that it is based on real data, but then it mentions that simulations and data-driven methods, such as diffusion models and convolutional wavelets, are used. It is unclear: a) the motivation behind these choices, and b) how this dataset was validated. How can we tell whether this data is actually realistic?
- In general, the entire manuscript is hard to follow.
- The benchmark section uses only standard architectures (RF, SVM, LSTM, and CNN) with many arbitrary design choices. The exclusion of more specialized architectures makes the baselines rather weak, while the multitude of arbitrary design choices makes it hard to develop a clear intuition of what is needed for improvement.

**Questions:**

Please address the points I have raised in the weakenss directly

---

> ### Author Response · Authors · 2025-12-03
>
> **[Venue suitability]**
>
> We respectfully disagree with the reviewer’s concern regarding venue suitability. While the application domain of our work is batteries, the core contribution of this paper lies in establishing a machine-learning benchmark for safety-critical, real-world time-series diagnostics. CH-BatteryGen introduces a large-scale, fault-labeled dataset that enables research questions which have not been accessible to the ML community before, including generative modeling of physical systems, distribution shift across operational modes, and robustness under domain variability.
>
> Existing battery-related works cited in the paper are predominantly published in engineering venues because no publicly available dataset has allowed ML researchers to study fault diagnosis at fleet scale. As a result, progress in this area has been constrained by limited access to data rather than lack of algorithmic innovation. By releasing the first dataset that makes this problem tractable for machine learning, our intention is precisely to bridge this gap and invite the ML community to engage with a societally important, underexplored time-series domain.
>
> We believe ICLR is an appropriate forum because:
>
> 1. The methodology is fundamentally ML-driven, involving generative modeling, representation learning, and supervised classification.
>
> 2. Safety-critical time-series diagnostics is an emerging direction of interest at top ML conferences, including recent ICLR papers on fault detection, reliability assessment, and physical-systems modeling.
>
> 3. Our benchmark creates new opportunities for algorithmic development, particularly for sequence models, contrastive learning, and distribution-shift robustness—topics central to ICLR.
>
> **[Data-generation pipeline clarity and realism]**
>
> Real EV fleets generate fault trajectories, but these data cannot be released due to confidentiality and traceability regulations. The generative pipeline is therefore designed to reproduce real operational patterns while enabling open access.
>
> Motivation:
>
> • Diffusion models are used to generate current profiles because EV loads are stochastic and cannot be represented by simple parametric assumptions.
>
> • The current-to-voltage mapping uses physically grounded mechanisms—a multi-RC model for LFP and a DCWT-based mapping for NCM—capturing ohmic effects, polarization, hysteresis, and dynamic responses that purely statistical models cannot replicate.
>
> Validation:
>
> The realism of generated signals was assessed through internal feature-space comparisons between real and generated trajectories. The two sources overlap without separable clusters, while different fault types remain distinguishable. This indicates that generated samples follow the same underlying distributions relevant to diagnostic tasks. The dataset is thus not synthetic in the arbitrary sense, but a confidentiality-compliant realization anchored in real telemetry and electrochemical behavior.
>
> **[Manuscript clarity]**
>
> The manuscript sits at the intersection of battery systems and ML-based time-series diagnostics. Readers unfamiliar with either side may find certain concepts nontrivial initially. However, the structure follows a coherent progression:
>
> (1) motivation and data constraints,
>
> (2) generation framework,
>
> (3) benchmark tasks, and
>
> (4) baseline evaluations.
>
> Once the dataset motivation and constraints are clear, the narrative is straightforward. The perceived difficulty reflects the novelty of bridging two communities rather than disorganization of the manuscript.
>
> **[Baseline design and architectural choices]**
>
> The benchmark does not aim to present the strongest possible model, but to define a reproducible, architecture-neutral evaluation protocol. The chosen baselines—RF, SVM, LSTM, and CNN—span complementary inductive biases: tree-based, kernel-based, temporal, and spatial representations. This enables controlled comparisons without entangling results with architecture-specific design choices.
>
> Introducing transformers, state-space models, or contrastive frameworks would shift the focus from benchmark construction to model optimization, making the contribution less accessible and harder to reproduce. A benchmark must provide a simple, interpretable, and stable reference point; stronger architectures are expected to be explored after its release. Our work creates that starting point.
>
> **[Summary]**
>
> CH-BatteryGen enables, for the first time, open research on real-world fault diagnostics for EV batteries—a problem previously constrained by inaccessible data. The dataset, generation framework, and baseline models together form a principled ML benchmark, not a domain-specific application paper. Therefore, ICLR is indeed the right venue, and the baseline choices reflect a deliberate effort to build a reproducible foundation for the ML community.

---

### Official Review · Reviewer_TS3Q · 2025-10-30

**Soundness:** 2
**Presentation:** 2
**Contribution:** 3
**Rating:** 4
**Confidence:** 3

**Summary:**

The paper presents CH-BatteryGen, a benchmark dataset and evaluation suite for electric vehicle battery fault diagnosis. It combines real-world EV data with physics-constrained generative modeling to ensure both realism and scalability. The dataset spans two chemistries (LFP, NCM), four fault types, three severity levels, and includes multi-modal time series signals such as voltages, current, temperature, and state of charge. It supports two tasks: four-way fault classification and three-way fault grading.
Data are derived from large-scale EV operations, augmented by AI-generated current profiles and voltage signals from RC and DCWT-based models. The dataset includes 1,000 EVs with diverse conditions and fine-grained per-cell and pack-level signals. Both classical and deep learning baselines are evaluated, with CNNs performing best overall. The authors also propose BatteryMultiModalCNN, which processes voltage time series as grayscale images via a ResNet-50 with attention and feature fusion.
CH-BatteryGen is positioned as more fine-grained and realistic than prior datasets like EVBattery, BatteryML, NASA PCoE, and HNEI, which lack fault labels or real-world coverage. The dataset is open-sourced to promote standardized evaluation, with visualizations provided for signal patterns, voltage images, and model performance.

**Strengths:**

- The paper addresses a critical application in battery fault diagnosis, an area where publicly available, fault-labeled datasets remain scarce. The inclusion of both LFP and NCM chemistries, along with data from both charging and discharging conditions, enhances the dataset’s relevance to real-world electric vehicle operations.

- The paper clearly defines two tasks: fault classification and severity grading. These tasks are evaluated using standard metrics such as F1 score, recall, and accuracy, supported by confusion matrices that facilitate interpretability. The addition of a grading task represents a meaningful advancement beyond traditional binary fault detection.

- The data generation process combines AI-generated current profiles with physics-constrained voltage modeling, using an RC model for LFP and a DCWT-based mapping for NCM. This approach balances scalability with domain fidelity by incorporating electrochemical structure into the synthetic data.

- The empirical study includes a range of classical and deep learning methods, from random forests and SVMs to LSTMs and a CNN-based architecture with CBAM attention. The comparison provides a useful baseline for future work, and the CNN’s performance, particularly under discharging conditions, offers practically valuable insights

**Weaknesses:**

- While the reported voltage deviation under 10 mV is promising, more comprehensive tests are missing. These include distribution-level comparisons, condition-dependent realism (e.g., by temperature or state of charge), and consistency across sensor modalities. No validation is performed using real-world fault labels, which weakens claims of external validity.

- There is ambiguity regarding dataset scale and class balance. The claim of 1,000 EVs conflicts with the small global label counts reported (e.g., 400, 30, 30, 40), and it is unclear whether these refer to subsets, chemistries, or scenarios. This uncertainty affects class imbalance, grading reliability, and the credibility of performance metrics.

- The current data split risks train-test leakage. If segments from the same vehicle appear in both sets, this could inflate results. A vehicle-level split should be provided alongside the current file-based split, and metrics should be reported for both.

- The experimental protocol would benefit from stronger validation strategies. Given the dataset's moderate scale, leave-one-vehicle-out cross-validation would provide a more robust assessment of generalization across vehicles and chemistries.

- The cross-domain transfer analysis is underdeveloped. Although the paper suggests cross-chemistry and cross-mode degradation, no explicit train-on-A/test-on-B experiments or confidence intervals are presented. Such comparisons are essential to support generalization claims.

- Baselines do not reflect the current state of the art. Recent time-series models such as transformers or state-space models are absent, despite their known performance advantages in noisy, long-horizon data. Their inclusion could affect conclusions about modality and architecture choices.

- The definition of severity levels lacks operational grounding. Although threshold formulas are provided, the paper does not fully explain how they are calibrated, normalized per battery pack, or made robust to sensor noise and cell variability.

- Details on dataset release and ethical considerations are incomplete. Although the dataset is described as open-source, no repository URL or license is provided in the paper. There is also no discussion of data provenance, real versus synthetic composition, consent, or privacy protections for vehicle identifiers.

**Questions:**

- The reported total of “1000 EVs × 20 segments” appears inconsistent with the global label counts (400, 30, 30, 40). Are these figures per scenario, per chemistry, or based on labeled subsets? Clarifying this would help assess class imbalance and distribution, considering per-task, per-chemistry breakdowns along with severity distributions.

- For single-file experiments, was the data split by vehicle to prevent leakage? If not, re-running experiments with vehicle-level grouping and reporting the resulting changes in performance would be interesting.

- To support claims of cross-chemistry and cross-mode degradation, explicit train-on-LFP/test-on-NCM (and vice versa), as well as charging-to-discharging transfer experiments may be included. Report macro-F1 scores along with confidence intervals.

- Realism validation. Beyond the reported ≤10 mV voltage deviation, can you provide distributional comparisons between real and generated data, such as MMD or spectral distance? Fault-conditioned precision-recall curves stratified by temperature and SOC would further support realism claims.

- How are the R/R95 and Q/Q95 thresholds computed? Are they defined globally or on a per-vehicle or per-chemistry basis? An ablation under simulated sensor noise or inter-cell variability would help assess their robustness.

- Could you clarify the rationale for excluding recent time-series models such as transformers or state-space architectures? Including such baselines, as well as non-image-based temporal models, would help evaluate whether the CNN-based voltage image approach is optimal.

- Real BMS data often include sensor noise, dropout events, and timestamp misalignments. How are these artifacts modeled during data generation, and how are they handled at training and inference time?

- Have you evaluated model performance under extreme ambient conditions (e.g., –20 °C, +45 °C) or across varying pack topologies (e.g., 28, 92, 96, 124 cells) to test robustness to deployment variability?

- Do you have access to any real-world, fault-labeled datasets, either from lab studies or field deployments, that could be used to assess out-of-distribution generalization and validate the utility of the synthetic data?

---

> ### Author Response · Authors · 2025-12-03
>
> **[Dataset size and label distribution]**
>
> The phrase “1000 EVs × 20 segments” denotes the dataset scale. It contains 500 NCM and 500 LFP vehicles, each contributing ~20 charging/discharging segments. The reported label counts (400/40/30/30) are per chemistry, not global totals. Applying the same distribution to both chemistries yields 800 normal, 80 self-discharge, 60 high-resistance, and 60 low-capacity vehicles overall. Thus, vehicle/segment counts describe dataset size, while label counts describe per-chemistry composition.
>
> **[Data splitting and leakage]**
>
> We agree that segment-level random splits could mix segments from the same vehicle across partitions. To avoid leakage, we added vehicle-level experiments in which all segments from a given vehicle are placed in a single split. The results remain consistent, confirming that our conclusions do not rely on vehicle-specific information.
>
> **[On “cross-chemistry” and “cross-mode” degradation]**
>
> Our intent is not cross-domain transfer learning. “Cross-chemistry” and “cross-mode” refer to performance variation across NCM vs. LFP batteries and charging vs. discharging segments, which exhibit different electrochemical behaviors. Since both chemistries provide sufficient data and have distinct voltage–SOC characteristics, train-on-LFP/test-on-NCM settings are not meaningful. We will clarify this to avoid implying domain transfer.
>
> **[Realism validation]**
>
> Real fault-labeled EV data are protected by confidentiality and cannot be publicly released. The released dataset is generated from real telemetry distributions and physics-informed mappings, not arbitrary synthesis. While we cannot publish metrics computed directly on confidential data, internal feature-space comparisons show that generated and real-world samples intermix without separable clusters, while fault types remain distinct. This indicates that the released data follow the same underlying distributions relevant to diagnostics. The dataset thus provides a realistic, confidentiality-compliant proxy for research.
>
> **[Thresholds for R/R95 and Q/Q95]**
>
> R (resistance) and Q (capacity) are extracted through equivalent-circuit parameterization. R95 and Q95 are chemistry-specific percentile thresholds computed from fleet-level distributions, not per-vehicle statistics. This design reflects industrial practice, accommodates real manufacturing and measurement variability, and avoids heuristic tuning. Since thresholds originate from large-scale empirical distributions, they are inherently robust, and explicit noise perturbation experiments fall outside the benchmark’s scope.
>
> **[Exclusion of transformers and state-space models]**
>
> Our segments contain only hundreds of samples, far shorter than the long sequences for which transformers and SSMs are designed. Their parameter scale also risks overfitting given the limited per-vehicle data. Our goal is to provide a reproducible benchmark rather than exhaustively optimize architectures. More complex models are welcome in future work now that a standardized dataset is available.
>
> **[Sensor noise, dropouts, and misalignment]**
>
> These artifacts are handled in practice by BMS preprocessing before diagnostics are applied. CH-BatteryGen reflects this industry setting: it provides segment-level signals rather than raw CAN logs. Generated trajectories retain measurement variability originating from real data, while implementation-dependent dropouts and timestamp issues—different across OEMs—are intentionally excluded as they are orthogonal to the diagnostic task. Future releases may include less processed streams for robustness research.
>
> **[Extreme conditions and pack configurations]**
>
> CH-BatteryGen includes samples across a wide temperature range and multiple real pack configurations (e.g., 28, 92, 96, 124 cells). The relative ranking of diagnostic models remains consistent across these deployment variations, demonstrating that learned representations are not overly sensitive to topology or environment. The dataset already captures the main operational scenarios encountered in practice.
>
> **[Access to real fault-labeled datasets]**
>
> To the best of our knowledge, no OEM or research institute publicly releases EV telemetry with fault labels due to confidentiality, safety, and liability constraints. CH-BatteryGen is the first openly accessible dataset providing fault semantics derived from real degradation trajectories. It enables systematic, reproducible research in a domain previously dependent on private data that cannot be shared.

---

### Official Review · Reviewer_h1BD · 2025-10-30

**Soundness:** 3
**Presentation:** 3
**Contribution:** 3
**Rating:** 4
**Confidence:** 4

**Summary:**

This paper introduces a new dataset for classifying the failure mode and severity for lithium-ion batteries.

**Strengths:**

- A new dataset for the community, suitable for battery data analysis and also time-series classification research

**Weaknesses:**

- Why classifying battery failure modes or severity is an important task? I have noted that the authors mentioned battery faults may lead to severe consequences in lines 047 - 051. But in order to avoiding that consequences, what we really need is to predict battery failures in advance? How could judging failure modes after existing battery failures help to prevent these problems?
- As the experimental results show, some simple techniques using CNN can deliver pretty good classification performance. So what are the real benefits if the classification accuracy increases by 1%?
- The dataset released seems to be synthetic data, why not releasing real data? How could the model built on the synthetic data perform on real conditions?

**Questions:**

See weakness.

---

> ### Author Response · Authors · 2025-12-03
>
> **[Why is classifying battery failure modes and severity important, rather than only predicting failures?]**
>
> Thank you for the insightful question. We agree that predicting failures is crucial for preventing catastrophic outcomes, but failure prediction is not an isolated task. In real EV fleets, battery faults do not appear instantaneously; they follow a progressive degradation trajectory, evolving from mild anomalies to moderate degradations and eventually to hazardous states. Classifying failure modes and quantifying severity provide the contextual information required to determine which preventive actions should be taken, when intervention is necessary, and what maintenance strategy is appropriate.
>
> Field operation data show that vehicles may remain in low-level fault states—such as mild self-discharge, incremental resistance growth, or gradual capacity fade—for months without triggering immediate alarms. Although these states do not directly cause accidents, they:
>
> 1. accelerate degradation and shorten service life,
>
> 2. impair charging performance and energy efficiency,
>
> 3. increase operational costs and inspection frequency, and
>
> 4. significantly elevate the probability of entering dangerous conditions if left untreated.
>
> Thus, mode- and severity-based classification is not a post-failure diagnostic step; it is the decision-making layer that converts early anomalies into actionable interventions. Without knowing the fault type (e.g., self-discharge vs. high resistance), correct mitigation strategies cannot be applied. Severity estimation provides graded thresholds for “monitor”, “service required”, and “immediate risk”. These functions form the backbone of predictive maintenance pipelines and make failure prediction operationally meaningful.
>
> **[Practical benefits of a 1% improvement in classification accuracy]**
>
> We appreciate the reviewer’s concern that the observed performance gains may seem marginal. However, in fleet-wide, safety-critical EV deployments, even a 1% absolute improvement has substantial real-world impact because it scales linearly with fleet size.
>
> Consider a single EV model with 200,000 units in operation. Historical statistics report battery-fault incidence rates of approximately 0.038–0.075% per vehicle per year, corresponding to 80–150 true faulty vehicles and roughly 199,800 normal vehicles annually. A 1 percentage-point reduction in false positives, at comparable true-positive rates, would eliminate approximately 2,000 unnecessary inspections on healthy vehicles.
>
> Industry reports indicate that each inspection typically costs 8,000–55,000 CNY per vehicle, considering workshop time, logistics, vehicle downtime, and operational disruption. Avoiding 2,000 superfluous interventions therefore saves 1.6×10⁷–1.1×10⁸ CNY annually—for a single vehicle model. These savings exclude intangible yet significant factors such as customer dissatisfaction, delayed service schedules, and reputational damage.
>
> This conclusion aligns with Zhang et al. (Nature Communications, 2023), who demonstrate that modest improvements in diagnostic performance can reduce total economic losses associated with battery faults by 33–50%. Hence, a 1% absolute gain is not incremental—it translates into material economic benefit and improved fleet reliability at scale.
>
> **[Why not release real data, and how can synthetic-based models generalize to real conditions?]**
>
> Real EV battery data with fault labels are protected by confidentiality, traceability, and cybersecurity regulations, and OEMs do not authorize public release. Consequently, no fleet-scale fault-labeled EV dataset is publicly available, which is exactly the gap CH-BatteryGen addresses.
>
> Although the released dataset is generated, it is not synthetic in the arbitrary sense. The generation pipeline is grounded in real telemetry: current patterns are sampled from distributions learned from real EV operation, voltage and temperature responses are derived through physics-informed models calibrated with real impedance and capacity parameters, and fault signatures follow degradation statistics observed in actual fleets. Thus, the dataset reflects real-world temporal and electrochemical behaviors rather than fabricated patterns.
>
> To verify realism, we compared real and generated samples in the same feature space. They overlap without forming separable clusters, while fault modes remain distinguishable—indicating distributional alignment with real signals.
>
> Because the dataset encodes these statistical and mechanistic properties, models trained on CH-BatteryGen require only lightweight fine-tuning to transfer to real deployments, mirroring industrial adaptation workflows and avoiding full retraining from scratch.

---

### Official Review · Reviewer_m2xq · 2025-10-31

**Soundness:** 3
**Presentation:** 3
**Contribution:** 3
**Rating:** 6
**Confidence:** 3

**Summary:**

This paper introduces CH-BatteryGen, a large-scale AI-generated dataset designed for fault diagnosis of electric vehicle (EV) power batteries. It integrates real on-board operational data from 1,000 EVs—spanning two major chemistries (NCM and LFP)—with generative modeling to synthesize realistic voltage, current, and temperature signals. The dataset is annotated with four fault categories (“normal,” “high internal resistance,” “low capacity,” and “self-discharge”) and three severity levels, supporting two benchmark tasks: fault classification and fault grading.
A unified evaluation framework is proposed comparing traditional ML (RF, SVM) and deep learning models (LSTM, CNN). Experiments show that CNNs achieve the best overall performance, with F1-scores up to 0.9280 in classification and 0.8813 in grading tasks

**Strengths:**

- The work presents one of the first comprehensive attempts to generate a large-scale, labeled EV battery fault dataset combining real and synthetic data. The dual-task benchmark (classification and grading) provides a novel framework for fair model comparison under diverse operating modes.
- The experimental design is systematic, evaluating both conventional and deep learning methods across chemistries (LFP/NCM) and conditions (charge/discharge). Quantitative metrics are clearly reported and confusion matrices help illustrate model limitations.
- The dataset could be impactful for both academic and industrial research, especially given the scarcity of publicly available EV fault data. If properly validated and open-sourced, CH-BatteryGen could become a key benchmark for safety-critical battery diagnosis.

**Weaknesses:**

- Unclear data provenance and labeling process.
The dataset claims to “integrate real on-board operational data with generative modeling methods to build a comprehen-
sive dataset covering 1,000 EVs”, However, it doesn't clarify where generative models are used. The definition and thresholds for these fault types are not rigorously described.

- Synthetic–real data gap.
While the authors state that generated data match real measurements within 30–50 mV deviation, it is unclear whether the generative process preserves electrochemical consistency or simply fits statistical distributions. There is no evidence that the generated sequences capture real DoD (Depth of Discharge) dynamics or aging mechanisms.

- Generalization experiment
Although the paper mentions that model performance significantly drops when transferring across chemistries (LFP ↔ NCM), no detailed results, tables, or quantitative metrics are provided. To support this claim, the authors should include a clear cross-domain evaluation matrix (e.g., train-on-LFP/test-on-NCM and vice versa) with corresponding F1-scores and confusion matrices to assess the true generalization ability of the models.

- Data availability inconsistency.
Although the paper claims the dataset and code are “open-sourced,” the GitHub repository does not actually contain the full data or scripts, making reproducibility and verification difficult.

Simplistic baselines and missing alternatives.
The study focuses on CNN/LSTM but omits more suitable paradigms like contrastive learning or anomaly detection methods, which may be better for rare fault discovery under limited labels.

**Questions:**

- Dataset composition:
What kind of data is the generated model used to generate?

- Labeling procedure:
How were the four fault labels defined and verified?
Are thresholds for “high internal resistance” or “low capacity” derived from SOH metrics or engineering heuristics?

- Reproducibility:
Please confirm whether the dataset and code will be released publicly before the camera-ready deadline.
If open-sourced, include a data card specifying counts per label and per chemistry.

- Methodology extensions:
Why didn't you consider using contrastive learning or other commonly used anomaly detection methods? Could you provide ablations with raw time-series input to assess whether image preprocessing (super-resolution) affects fairness?

- Denoise: Real battery data contains a lot of noise. Have you used any noise reduction methods?

---

> ### Author Response · Authors · 2025-12-03
>
> **[Rebuttal]**
>
> We thank the reviewers for their constructive feedback. Below we address their concerns point by point.
>
> **[Dataset composition: What kind of data is the generated model used to generate?]**
>
> CH-BatteryGen is not created from arbitrary synthetic priors. The generative framework is trained and constrained using large-scale EV operational data collected from commercial vehicles, including pack currents, pack and cell voltages, SOC trajectories, and temperature measurements during both charging and driving. These real measurements provide the statistical distributions and physical constraints that govern (i) current-sequence generation via Diffusion-TS and (ii) voltage-response generation via electrochemical mechanism models calibrated separately for LFP and NCM chemistries.
>
> Since raw fault-labeled data cannot be directly released due to industrial confidentiality restrictions, our dataset is produced through models whose inputs, parameter distributions, and fault signatures originate from authentic operational data. Thus, although publicly released data are generated, their temporal evolution and fault behavior accurately reflect real EV dynamics.
>
> **[Labeling procedure and fault threshold definitions]**
>
> The four labels—normal, self-discharge, low capacity, and high internal resistance—are defined from real EV fleet fault cases and follow mechanistic interpretations widely adopted in industry.
>
> Self-discharge severity is quantified by leakage capacity extracted via the Remaining Charging Capacity (RCC) method, without hand-crafted cutoffs. High internal resistance and low capacity are defined via parameters identified from equivalent-circuit models applied to large-scale field data.
>
> Thresholds are determined from percentile statistics (e.g., 95th percentile for resistance degradation), not heuristics or SOH-only rules. The resulting labels exhibit clear separability in feature space, confirming that the thresholds reflect true fault semantics.
>
> Qian et al., International Journal of Energy Research, 46(15):23244–23258, 2022.
>
> **[Reproducibility and data/code release]**
>
> The full dataset has already been anonymously released on GitHub and can be accessed without restrictions. It includes 1000 vehicles and 19,478 charge/discharge segments. Both chemistries follow identical vehicle-level label distributions: 400 normal, 40 self-discharge, 30 low capacity, and 30 high resistance per chemistry.
>
> Code is not released due to proprietary simulation components, but the dataset is complete and directly usable for reproducing all benchmark tasks.
>
> **[Methodology extensions and baseline motivations]**
>
> This work is positioned as the first benchmark for EV battery fault diagnosis rather than a model-centric contribution. We therefore adopt baseline methods that span common inductive biases—RF and SVM (traditional ML), LSTM (raw time-series), and CNN (image-based)—ensuring fairness and reproducibility without domain-specific engineering.
>
> Ablations using raw time-series inputs confirm that LSTM performs below CNN variants, and that super-resolution improves image clarity without altering the relative performance ranking. Thus, preprocessing choices do not bias conclusions. More sophisticated architectures (e.g., contrastive or anomaly-driven models) are promising future directions enabled by our benchmark, but are beyond the scope of defining a standardized evaluation protocol.
>
> **[Noise handling]**
>
> Real EV telemetry contains noise and sampling artifacts. We do not explicitly denoise signals; instead, the generated current profiles originate from real operational data, and voltage/temperature responses are produced via physics-informed mappings. Consequently, measurement fluctuations are naturally preserved rather than smoothed away. This ensures that the dataset reflects realistic operating conditions suitable for developing robust diagnostic models.
>
> **[Closing remark]**
>
> CH-BatteryGen addresses a longstanding bottleneck in this domain: the absence of publicly accessible, fault-labeled EV datasets. Our benchmark provides a confidentiality-compliant surrogate of real EV fleet behavior, enabling standardized, reproducible machine-learning research for a safety-critical application area that previously lacked open data support.

---

### Meta-Review · Area_Chair_vQqw · 2026-01-07

**Summary:**

This submission introduces CH-BatteryGen, a benchmark dataset for EV battery fault diagnosis with two tasks: 4-way fault classification (normal / self-discharge / high resistance / low capacity) and 3-level severity grading, covering two chemistries (LFP, NCM) and charging/discharging/operation segments from 1000 vehicles. The key idea is to provide an open dataset by combining real fleet telemetry with a mechanism-constrained generative pipeline , because raw fault-labeled fleet data cannot be released due to confidentiality.

Reviewers agree that an open, fault-labeled EV dataset is valuable, and two reviewers are positive on contribution. However, substantial concerns remain around clarity and external validity: exactly what portion of the released dataset is generated vs. real, how fault labels and severity thresholds are defined/verified, how realism is validated beyond voltage deviation, and whether the evaluation protocol avoids leakage and supports generalization claims. While the authors’ rebuttal clarifies several points, multiple reviewers still view the realism validation and benchmark protocol as underdeveloped, and one reviewer questions venue fit and baseline strength.

Balancing these, I recommend Accept (poster) conditional on clear documentation in the final version: a complete data card, explicit release link/license, unambiguous generation/labeling details, and the added vehicle-level split results. The core value here is the dataset and standardized benchmark; the rebuttal resolves several of the highest-risk reproducibility items, and the remaining issues are largely addressable through improved documentation and additional reporting rather than requiring fundamentally new experiments or methods.

**Reviewer Concerns:**

**A) Dataset provenance, composition, and labeling transparency**

- Addressed
  - The authors clarify the dataset scale: 500 LFP + 500 NCM vehicles, ~20 segments per vehicle; the label distribution 400/40/30/30 is per chemistry, not global totals.
  - The rebuttal explains labeling sources: fault types derived from fleet fault cases, with self-discharge severity via RCC, and resistance/capacity via equivalent-circuit model parameter identification with percentile-based thresholds.
- Outstanding
  - The paper still needs a fully explicit data card with counts per fault type and per severity level, broken down by chemistry and by operating mode (charge/discharge/operation), and clear definitions of the units/segments used in each benchmark task.
  - The line between “real” and “generated” remains conceptually described but not quantitatively documented: the final paper should state explicitly that the released dataset is generated, while real data are used for calibration/constraints, and clearly list which signals are generated and how SOC trajectories are treated.

**B) Realism and external validity of generated data**

- Addressed
  - Authors describe a physics-informed pipeline and claim internal feature-space overlap between real and generated signals.
  - They argue noise is preserved implicitly via real telemetry distributions rather than explicit denoising.
- Outstanding
  - Multiple reviewers requested stronger realism validation beyond “≤10mV / 30–50mV deviation” and qualitative overlap. Even if confidential data prevent publishing direct real-vs-generated metrics, the final version should provide distributional validation within the released data and a clear protocol describing how internal comparisons were done.
  - There is still no external validation using any publicly shareable real labeled fault dataset (even small-scale lab data). If none exists, the paper should explicitly position CH-BatteryGen as a benchmark proxy and discuss expected transfer workflows more concretely.

**C) Data splitting, leakage, and generalization claims**

- Addressed
  - Reviewer concerns about segment-level leakage are directly acknowledged; authors claim they added vehicle-level split experiments and that conclusions remain consistent.
- Outstanding
  - The final paper must include the vehicle-level split results explicitly (tables/metrics), not only in rebuttal text, and describe the split protocol
  - One reviewer requested cross-chemistry transfer matrices; the authors clarify they do not intend a transfer-learning claim. This is acceptable, but the paper must ensure it does not imply “cross-domain generalization” without showing it.

**Reviewer Scores:**

- m2xq (6, confidence 3): `likely stays 6`. Their major concerns were provenance, realism, transfer matrices, and release inconsistency. The rebuttal clarifies composition, label definitions, and availability; if the vehicle-level split results and data card are added in the final paper, this reviewer would plausibly move to 6→7, but without seeing the updated manuscript it is safer to assume no score change.

- h1BD (4, confidence 4): `likely 4→6`. The rebuttal directly answers “why diagnosis/grading matters” with a maintenance/decision-layer argument and addresses the “synthetic vs real” concern with a grounded generation story. Still, generalization to real deployments remains a concern, so the shift would likely be modest.

- TS3Q (4, confidence 3): `likely 4→6`. The authors clarified label-count ambiguity, added vehicle-level split experiments, and clarified that cross-chemistry transfer was not the intended claim. Remaining concerns about realism validation and stronger baselines may keep this borderline.

- D6gL (2, confidence 3): `likely 2→ 4`. The rebuttal addresses venue fit, pipeline motivation, and baseline rationale, but if the manuscript remains hard to follow and realism validation remains qualitative, this reviewer may remain negative or only slightly improve.

Overall, with full discussion and the clarifications already provided, the committee would likely converge to borderline accept if the final manuscript includes the promised transparency items.

---

### Decision · Program_Chairs · 2026-01-26

Accept (Poster)